# Neural Program Search: Solving Programming Tasks from Description and Examples

## Abstract

We present a Neural Program Search, an algorithm to generate programs from natural language description and a small number of input / output examples. The algorithm combines methods from Deep Learning and Program Synthesis fields by designing rich domain-specific language (DSL) and defining efficient search algorithm guided by a Seq2Tree model on it. To evaluate the quality of the approach we also present a semi-synthetic dataset of descriptions with test examples and corresponding programs. We show that our algorithm significantly outperforms sequence-to-sequence model with attention baseline.

## 1 Introduction

The ability to synthesize a program from user intent (specification) is considered as one of the central problems in artificial intelligence (Green (1969)). Significant progress has been made recently in both program synthesis from examples (e.g. Balog et al. (2016), Polozov & Gulwani (2015), Ellis & Gulwani (2017)) and program synthesis from descriptions (e.g. Desai et al. (2016), Zhong et al. (2017), Lin et al. (2017), Ling et al. (2016)).

Programming by example techniques such as Flash Fill (Gulwani et al. (2012)) and BlinkFill (Singh (2016)) were developed to help users perform data transformation tasks using examples instead of writing programs. These methods rely on a small domain-specific language (DSL) and then develop algorithms to efficiently search the space of programs. Two shortcomings of these approaches are that DSL limits types of programs that can be synthesized, and that large engineering effort is needed to fine-tune such systems.

Program synthesis from description has not been applied widely in practice yet. One of the challenges is that the natural language is very ambiguous, yet there are very strict requirements for the synthesized programs (see Yin & Neubig (2017) and Rabinovich et al. (2017) for some discussion). In this paper we present Neural Program Search that learns from both description and examples and has high accuracy and speed to be applicable in practice.

We specifically consider a problem of synthesizing programs from a short description and several input / output pairs. By combining description and sample tests we address both limitations of programming by example and natural language program inference. We propose LISP-inspired DSL that is capable of representing solutions to many simple problems similar to those given as data transformation homework assignments, but is rather concise, making it more tractable to search in the space of programs in this DSL.

We propose a combination of two techniques — search in the programs space that is guided by a deep learning model. This way we can use the latest advances in natural language understanding with the precision of the search techniques. We use a Seq2Tree model (Alvarez-Melis & Jaakkola (2016)) that consists of a sequence encoder that reads the problem statement and a tree decoder augmented with attention that computes probabilities of each symbol in an AST tree node one node at a time. We then run a tree beam search that uses those probabilities to compute a number of most likely trees, and chooses one that is consistent with the given input/output pairs.

To evaluate the proposed model we have created a partially synthetic dataset AlgoLISP consisting of problem statements, solutions in our DSL and tests. We show that search guided by deep learning models achieves significantly better results than either of the two techniques separately.

## 2 RELATED WORK

We describe the related work from domains of programming by example, programming from description, latent program induction and related field of semantic parsing.

**Programming by Example**    There were several practical applications of programming by example based on search techniques and carefully crafted heuristics, such as Gulwani (2014). Some recent work that incorporates deep learning models into traditional search techniques includes DeepCoder (Balog et al. (2016)) and Deep API Programmer (Bhupatiraju et al. (2017)). Gaunt et al. (2016) provides a comparison of various program synthesis from examples approaches on different benchmarks, showing limitations of existing gradient descent models.

**Programming from Description**    Program synthesis from natural language descriptions has seen revival recently with progress in natural language understanding, examples of such work include Desai et al. (2016), Zhong et al. (2017), Lin et al. (2017) and Ling et al. (2016). Advances in this field are limited by small and/or noisy datasets and limitation of existing deep learning models when it comes to decoding highly structured sequences such as programs.

**Latent Program Induction**    There has been a plethora of recent work in teaching neural networks the functional behavior of programs by augmenting the neural networks with additional computational modules such as Neural Turing Machines (Graves et al. (2014)), Neural GPUs (Kaiser & Sutskever (2015)), stacks-augmented RNNs (Joulin & Mikolov (2015)) and Neural Program-Interpreters (Reed & De Freitas (2015)). Two main limitations of these approaches are that these models must be trained separately for each task and that they do not expose interpretable program back to the user.

**Semantic Parsing**    Semantic parsing is a related field to program synthesis from description, in which the space of programs is limited to some structured form. Noticeable work includes Dong & Lapata (2016) and Berant et al. (2013). In another line of research latent programs for semantic parsing are learned from examples, e.g. Neelakantan et al. (2016), Liang et al. (2016).

## 3 NEURAL PROGRAM SEARCH

This section describes the DSL used for modeling, out neural network architecture and an algorithm for searching in program space.

### 3.1 DOMAIN SPECIFIC LANGUAGE

There are multiple reasons to use a domain specific language for code generation instead of an existing programming language. One reason is to be able to convert a program to multiple target languages for practical applications (e.g. SQL, Python, Java), which requires our DSL to be sufficiently general. Second, designing a DSL from scratch allows to add constrains that would simplify its automated generation.

Our DSL is inspired by LISP – functional language that can be easily represented as an Abstract Syntax Tree and supports high-order functions. We augmented our DSL with a type system. While types do not appear in programs, each constant, argument or function has a type. A type is either an integer, a string, a boolean, a function or an array of other non-function types.

A program in the DSL comprises a set of arguments (where each argument is defined by its name and type) and a program tree where each node belongs to one of the following symbol types: **constant**, **argument**, **function call**, **function**, or **lambda**. See Figure 1 for a partial specification of the DSL.

The DSL also has a library of standard functions. Each function has a return type and a constant number of arguments, with each argument having its own type. The type system greatly reduces the number of possible combinations for each node in the program tree during search.

```
program ::= symbol
symbol ::= constant | argument | function_call | function | lambda
constant ::= number | string | TRUE | FALSE
function_call ::= (function_name arguments)
function ::= function_name
arguments ::= symbol | arguments , symbol
function_name ::= REDUCE | FILTER | MAP | HEAD | + | - ...
lambda ::= LAMBDA function_call
```

Figure 1: Partial specification of the DSL used for this work.

## 3.2 SEQ2TREE

Our neural network model uses an attentional encoder-decoder architecture. The encoder uses RNN to embed concatenation of arguments ARGS and tokenized textual description of the task TEXT. The decoder is a doubly-recurrent neural network for generating tree structured output (Alvarez-Melis & Jaakkola (2016)). At each step of decoding, attention is used to augment current step with relevant information from encoder.

Formally, let $T = \{V, E, L\}$ be connected labeled tree, where $V$ is the set of nodes, $E$ is set of edges and $L$ are node labels. Let $H^e$ be a matrix of stacked problem statement encodings (outputs from encoder's RNN). Let the $g^p$ and $g^s$ be functions which apply one step of the two separate RNNs. For a node $i$ with parent $p(i)$ and previous sibling $s(i)$, the ancestral and fraternal hidden states are updated via:

$$c_i^p = context(x_p(i), H^e) \qquad h_i^p = g^p(h_p^p(i), c_i^p) \qquad (1)$$

$$c_i^s = context(x_s(i), H^e) \qquad h_i^s = g^s(h_s^s(i), c_i^s) \qquad (2)$$

where $x_p(i)$, $x_s(i)$ are the vectors representing the previous siblings and parents values, respectively. And $context(x, H^e)$ computes current context using general attention mechanism (Luong et al. (2015)) to align with encoder presentations using previous parent or sibling representation and combining it with $x$ in a non-linear way:

$$a = softmax(H^e W_a x) \qquad (3)$$

$$r = a^T H^e \qquad (4)$$

$$context = tanh((r\|x)W_c) \qquad (5)$$

where $\|$ indicates vector concatenation and $W_a$ and $W_c$ are learnable parameters. Once the hidden depth and width states have been updated with these observed labels, they are combined to obtain a full hidden state:

$$h_i = (U^p h_i^p + U^s h_i^s) \qquad (6)$$

where $U_p$ and $U_s$ learnable parameters. This state contains combined information from parent and siblings as well as attention to encoder representation and is used to predict label of the node. In a simplest form (without placeholders), the label for node $i$ can be computed by sampling from distribution:

$$o_i = softmax(W h_i) \qquad (7)$$

After the node's output symbol $\hat{l}_i$ has been obtained by sampling from $o_i$, $x_i$ is obtained by embedding $\hat{l}_i$ using $W^T$. Then the cell passes $(h_i^p, x_i)$ to all it's children and $(h_i^s, x_i)$ to the next sibling (if any), enabling them to apply Eqs (1) and (2) to compute their states. This procedure continues recursively following schema defined by DSL that is being decoded.

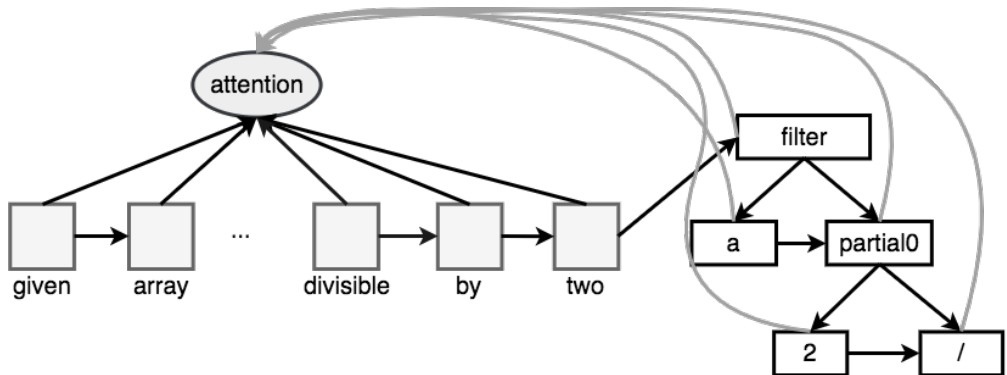

Figure 2: Example of SEQ2TREE encoder-decoder model for "given an array, return values divisible by two". Left part is an encoder with embeddings+GRU cell, right is doubly-recurrent decoder with attention.

The model is trained using back-propagation. The teacher forcing is used by using target topology of the code tree, feeding target labels for parent / sibling nodes. The error is obtained using the cross-entropy loss of $o_i$ with respect to the true label $l_i$ for each decoded node.

Exploring alternative methods of training, such as REINFORCE (similar to Zhong et al. (2017)) or using SEARCH at training time is left for future work.

### 3.3 SEARCH

One of the central ideas of this work is to use Tree-Beam search in the program space using a deep learning model to score symbols in each AST node. The search continues until a complete program is found that passes given sample input / output pairs.

Search algorithm described in Algorithm 1 starts with a priority queue with a single empty program. At all times, we only keep the top QUEUE_N most probable trees built so far in the priority queue.

---

**Algorithm 1** Tree-Beam Search

---

1:  $queue \leftarrow$ HeapCreate()
2:  $model \leftarrow$ Seq2Tree($task\_description$)
3:  $trees\_visited \leftarrow 0$
4:  HeapPush($queue$, EMPTY_TREE)
5:  **while** HeapLength($queue$) > 0 **and** $trees\_visited$ < MAX_VISITED **do**
6:      $cur\_tree \leftarrow$ HeapPopMax($queue$)
7:      $empty\_node \leftarrow$ FindFirstEmptyNode($cur\_tree$)
8:      **if** $empty\_node$ = null **then**
9:          $trees\_visited \leftarrow trees\_visited + 1$
10:         **if** RunTests($cur\_tree$, $sample\_tests$) = PASS **then return** $cur\_tree$
11:         **else   continue**
12:     **for all** ($prob$, $symbol$) in GetProbs($model$, $empty\_node$) **do**          ▷ In decreasing order of probabilities
13:         **if** $prob$ < THRESHOLD **then   break**
14:         **if** SymbolMatchesNodeType($symbol$, $empty\_node$) **then**
15:             $new\_tree \leftarrow$ CloneTreeAndSubstitute($cur\_tree$, $empty\_node$, $symbol$)
16:             HeapPush($queue$, $new\_tree$)
17:     **while** HeapLength($queue$) > QUEUE_N **do**
18:         HeapPopMin($queue$)
19: **return** null

---

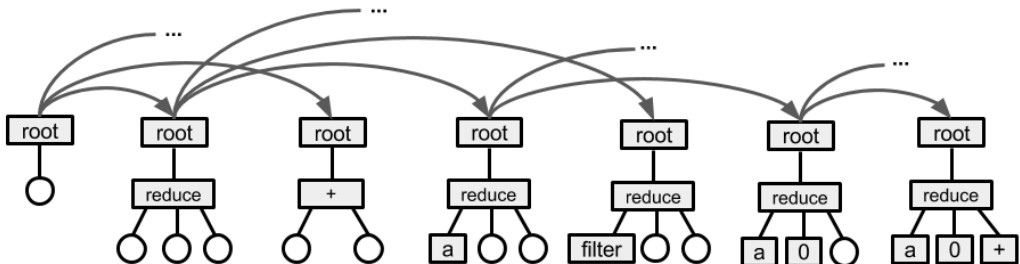

Figure 3: Example of tree search for a query "Given an array, find the sum of its elements". Rectangles represent nodes with a symbol, while circles represent empty nodes. We start with an empty tree on the far left. When that tree is popped from the priority queue, we consider each possible symbol for the first empty node in the pre-order traversal, and create a new tree for each. Two such trees are shown in this figure, for symbols *reduce* and +. When the tree with *reduce* is popped, several new trees are generated by filling in the first empty node in the pre-order traversal of that tree, which is the first child of reduce. The first argument of reduce is an array, so only symbols that produce arrays are considered. Two trees for such symbols are shown on the figure for *a*, which is an argument, and *filter*. The search continues until either D trees are generated, or a tree that passes all the sample tests is found. Such tree is shown on the far right.

If a program on the top of the queue is complete (no more nodes need to be added), we run evaluation with given sample input / output examples. If the results from current program match expected outputs, search is stopped. Alternatively, if over MAX_VISITED programs has already been evaluated, the search stops without program found.

Each program in the priority queue is represented as an incomplete tree with some nodes already synthesized and some still empty. When such incomplete tree $T$ is popped from the queue, we locate the first empty node $n$ in the pre-order traversal of the tree, and use SEQ2TREE model to compute probabilities of each possible symbol being in that node. At that point we already know the type of the symbol the node should contain, and thus only consider symbols of that type. For each such symbol $s$ we construct a new tree by replacing $n$ with $s$. We then push all the new trees, no matter how unlikely they are into the priority queue, and then remove least probable trees until the size of the priority queue is not QUEUE_N or less.

In our experiments evaluating the SEQ2TREE model takes comparable amount of time to cloning trees and pushing them to the queue, so optimizing both steps would contribute to the performance of the search. We use the following optimization techniques:

**Persistent trees**. After evaluating the model once we need to clone the tree as many times as many symbols will be considered for the first empty node. Storing all the trees in full can be memory consuming, and, depending on the language in which the search is implemented, allocating objects for the nodes can take considerable amount of time. One way to save memory and time on cloning trees is to use persistent trees. When a new tree $T_{new}$ is created from a tree $T$ by introducing a new node $s$, it is sufficient to clone only the nodes on the path from root to $s$, and replace their corresponding children on that path with the cloned version. This takes time and memory proportional to the height of the tree, which for larger trees is significantly smaller than the total number of nodes in the tree. The tree is then represented as a pointer to the root node.

**Batched search**. During training we need to read trees of different shapes, which is a challenging problem, and we use dynamic batching to address it. During search we only invoke SEQ2TREE on a single node, so multiple such invocations can be trivially batched. We batch invocations of the SEQ2TREE across tasks in the following way: we run the search for *batch_size* tasks simultaneously, and on each step pop the single most likely incomplete tree for each task, identify the empty node in each of them, and compute the probabilities of the symbols in all of them at once.

Table 1: ALGOLISP statistics.

|  | Train | Dev | Test |
|---|---|---|---|
| # tasks | 79, 708 | 9, 352 | 10, 940 |
| Avg text len | 38.72 | 39.95 | 37.58 |
| Avg code depth | 8.12 | 8.23 | 7.97 |
| Avg code len | 28.31 | 29.31 | 27.16 |
| Vocab size | | 230 | |

This approach speeds up evaluation of the search if it is run on multiple tasks simultaneously, for example when evaluating the accuracy on a held out set. However, in cases when only one task is being evaluated batching across tasks is not applicable. We evaluated the following alternative: on each iteration of search pop the top *batch_size* incomplete trees from the priority queue instead of just the single most likely one, identify the empty node in each of them, and compute the probabilities of symbols in all of them at once. This approach did not produce any noticeable speed up, and in most cases even slowed the search down slightly. The possible reason for that is that if the model that guides the search is good, the correct incomplete tree will be the top one most of the time, so number of model evaluations that are saved due to batched execution is very small, and the extra computation time of evaluating a model on a batch instead of a single sample outweighs the time saved due to those few extra evaluations.

## 4 ALGOLISP

In this section we describe a new dataset we prepared to train and evaluate models that learn to synthesize simple data processing programs.

ALGOLISP is a dataset of problem descriptions, corresponding implementations of the problem in a LISP-inspired programming language described in section 3.1 and tests. Each problem has 10 tests, where each test is input to be fed into the synthesized program and the expected output the program should produce. All the problems are designed in such way that the output for each input is unique. See Table 1 for dataset statistics.

There are multiple existing datasets for code synthesis task from natural language. Some recent notable ones are description to bash command (Lin et al. (2017)) and description to SQL (Zhong et al. (2017)). To the best of our knowledge, no existing dataset is applicable to our problem due to reasons such as no easy way of evaluating the results, insufficient complexity of the programs or size too small for deep learning.

Because same problem can be solved with many different programs, the solution is considered correct if it produces correct output on all the tests for given problem. For consistency and comparable results we suggest two specific approaches in which the tests are used during inference: no tests using at inference time (used for deep learning only models) and using first 3 tests for search and the remaining 7 tests as a holdout to evaluate correctness of the found program.

The dataset was synthesized with the following procedure (see Table 2 for examples). We first chose several dozen tasks from homework assignments for basic computer science and algorithms courses. For each task, we parameterized assignments (e.g. in statement "find all even elements in an array" *even* could be replaced by {prime, even, odd, divisible by three, positive, negative})and matching code. The final dataset is then random combination of such tasks, where other tasks can be passed into the given statement as input (e.g. two statements "find all even elements in an array" and "sort an array" will be combined to "find all even elements in an array and return them in sorted order").

This dataset is designed for the task of learning basic composition and learning to use simple concepts and routines in the DSL. Due to the fact that the number of homework assignments used for this dataset was relatively low, it is unlikely that the models trained on this dataset would generalize to new types of algorithm.

Table 2: Examples from ALGOLISP dataset. First row is an example of user provided homework assignment with program in our DSL. Subsequent lines are examples of synthesized tasks and programs, showing various properties of the generator: different text for the task, combination with other sub-problems (such as "elements in *a* that are present in *b*") and variation of task properties.

| | |
|---|---|
| You are given an array *a*. Find the smallest, element in *a*, which is strictly greater than the minimum element in *a*. | `(reduce (filter a`
`  (partial0 (reduce a inf) <))`
`  inf min)` |
| **Synthesized examples** ||
| Consider an array of numbers *a*, your task is to compute largest element among values in *a*, which is strictly smaller than the maximum element among values in *a*. | `(reduce (filter a`
`  (partial0 (reduce a -inf) >))`
`  -inf max)` |
| Given arrays of numbers *a* and *b*, compute largest element among elements in *a* that are present in *b*, which is strictly less than maximum element among elements in *a* that are present in *b*. | `(reduce (filter`
`  (filter a (partial0 b contains))`
`  (partial0 (reduce`
`    (filter a (partial0 b contains))`
`    inf) <))`
`  inf min)` |
| Given an array of numbers, your task is to find largest element among values in the given array that are divisible by two, which is strictly less than maximum element among values in the given array that are divisible by two. | `(reduce (filter`
`  (filter a is_odd)`
`  (partial0 (reduce`
`    (filter a is_odd)`
`    -inf) >))`
`  -inf max)` |

To make sure that the models are learning to compose simpler concepts for novel problems, the dataset split into train, dev, and test by surface form of the code. Thus ensuring that at training time the model has not observed any programs it will be evaluated on.

To evaluate neural networks and search driven algorithms, we compare output of the generated programs on a holdout set of tests for each task. Thus accuracy on this dataset is defined as Acc = $\frac{N_C}{N}$, where N is total number of tasks and $N_C$ is number of tasks for which the synthesized solution passes all the holdout tests.

## 5 EXPERIMENTS

We implemented all models using PyTorch[1] and used Dynamic Batching (e.g. Neubig et al. (2017)) to implement batched tree decoding at training time. We train using ADAM (Kingma & Ba (2014)), embedding and recurrent layers have hidden size of 100 units.

The placeholders are used to handle OOV (Hewlett et al. (2016)) in all neural networks. Placeholders are added to the vocabulary, increasing the vocabulary size from $N_v$ to $N_v + N_p$, where $N_p$ is a fixed size number of placeholders, selected to be larger than number of tokens in the input. The same OOV tokens from inputs and outputs are mapped to the same placeholder (selected at random from not used yet), allowing model to attend and generate them at decoding time. Given the attention mechanism this is very similar to Pointer Networks (Vinyals et al. (2015)).

### 5.1 RESULTS

We compare our model with Attentional Sequence to Sequence similar to Luong et al. (2015). Sequence to sequence models have shown near state of the art results at machine translation, question answering and semantic parsing.

The Table 3 presents results on ALGOLISP dataset for SEQ2SEQ+ATT and SEQ2TREE model with and without applying search described in section 3.3. Additionally performance of the SEARCH on its

---

[1]http://pytorch.org

Table 3: Performance on ALGOLISP. Accuracy is defined in section 4.

| Model | Dev Acc | Test Acc |
|---|---|---|
| Attentional Seq2Seq | 54.4% | 54.1% |
| Seq2Tree | 61.2% | 61.0% |
| Search | 0.5% | 0.6% |
| **Seq2Tree + Search** | **86.1%** | **85.8%** |

own is presented, to show result of search through program space without machine learning model guidance by only validating on input / output examples.

Explicitly modeling tree structure of code in SEQ2TREE improves upon attentional sequence to sequence model by 11%. SEARCH on it's own finds very limited number of programs with the same limit $MAX\_VISITED = 100$ (see 5.2 for details) as SEQ2TREE + SEARCH. Final model SEQ2TREE + SEARCH combines both approaches into one model and improves to the best result – 90.1%.

## 5.2 ANALYSIS

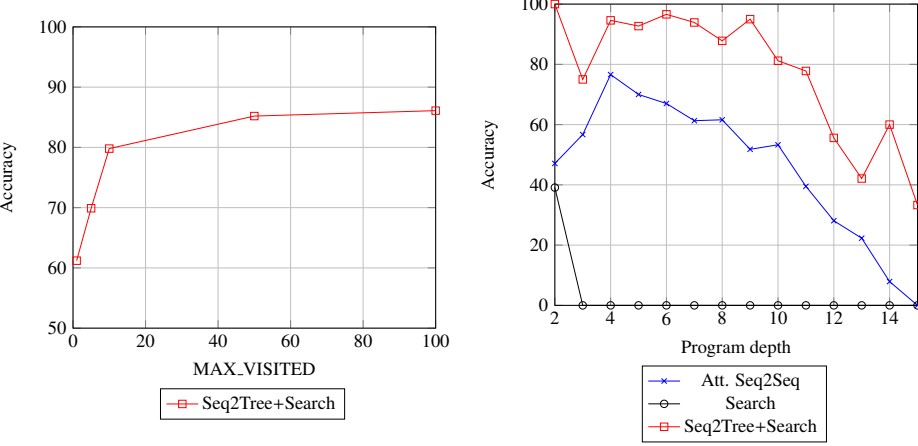

Figure 4: Analysis of results on dev set. Left plot shows accuracy of the model varying MAX_VISITED in SEARCH algorithm. Right plot shows accuracy stratified by depth of the target code tree.

**The pattern of how accuracy changes with the number of trees visited during search shows the quality of the neural network**. In general, given no limit on $MAX\_VISITED$, SEARCH will explore the entirety of the program space and find all programs that solve sample tests, which in our case contains on the order of $10^{2^D}$ programs, where $D$ is depth of programs explored. To compare improvement that neural network model brings to search, we compare the model performance at different thresholds of $MAX\_VISITED$. See Figure 4 for results. As expected, the accuracy of the model grows if the search gets to explore more trees. Interestingly, the growth of accuracy of SEQ2TREE+SEARCH slows down very quickly. It is expected if the neural network is good, since then it predicts correct symbols with high accuracy, and therefore the correct tree is more likely to be found early during the search.

**Depth of the program is a reasonable proxy for complexity of the problem**. Right part of Figure 4 shows accuracy of the models based on gold program depth. Note that there are relatively few programs with depth below 5 in the dev set, which leads to higher variance. As expected, with the growth of the depth of the tree, the accuracy reduces, since more nodes need to be predicted.

## 6 CONCLUSION

We have presented an algorithm for program synthesis from textual specification and a sample of input / output pairs, that combines deep learning network for understanding language and general programming patterns with conventional search technique that allows to find correct program in discrete space which neural models struggle with. We presented a semi-synthetic dataset to empirically evaluate learning of program composition and usage of programing constructs. Our empirical results show improvement using combination of structured tree decoding and search over attentional sequence to sequence model.

There remain some limitations, however. Our training data currently is semi-generated and contains only limited set of types of problems. It is prohibitively expensive to collect a human annotated set with large quantity of tasks per each problem type, so finding a way to learn from few examples per problem type is crucial. Additionally, in many practical use cases there will be no input / output examples, requiring interaction with the user to resolve ambiguity and improved techniques for structural output decoding in neural networks.

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
