# OpenReview forum: "Neural Program Search: Solving Data Processing Tasks from Description and Examples"
_ICLR.cc/2018/Conference — Invite to Workshop Track_

### Official Review · AnonReviewer2 · 2017-11-25
**Promising direction, but too preliminary**

**Rating:** 4
**Confidence:** 4

**Review:**

This paper presents a seq2Tree model to translate a problem statement in natural
language to the corresponding functional program in a DSL. The model uses
an RNN encoder to encode the problem statement and uses an attention-based
doubly recurrent network for generating tree-structured output. The learnt model is
then used to perform Tree-beam search using a search algorithm that searches
for different completion of trees based on node types. The evaluation is performed
on a synthetic dataset and shows improvements over seq2seq baseline approach.

Overall, this paper tackles an important problem of learning programs from
natural language and input-output example specifications. Unlike previous
neural program synthesis approaches that consider only one of the specification
mechanisms (examples or natural language), this paper considers both of them
simultaneously. However, there are several issues both in the approach and the
current preliminary evaluation, which unfortunately leads me to a reject score,
but the general idea of combining different specifications is quite promising.

First, the paper does not compare against a very similar approach of Parisotto et al.
Neuro-symbolic Program Synthesis (ICLR 2017) that uses a similar R3NN network
for generating the program tree incrementally by decoding one node at a time.
Can the authors comment on the similarity/differences between the approaches?
Would it be possible to empirically evaluate how the R3NN performs on this dataset?

Second, it seems that the current model does not use the input-output examples at
all for training the model. The examples are only used during the search algorithm.
Several previous neural program synthesis approaches (DeepCoder (ICLR 2017),
RobustFill (ICML 2017)) have shown that encoding the examples can help guide
the decoder to perform efficient search. It would be good to possibly add another
encoder network to see if encoding the examples as well help improve the accuracy.

Similar to the previous point, it would also be good to evaluate the usefulness of
encoding the problem statement by comparing the final model against a model in which
the encoder only encodes the input-output examples.

Finally, there is also an issue with the synthetic evaluation dataset. Since the
problem descriptions are generated syntactically using a template based approach,
the improvements in accuracy might come directly from learning the training templates
instead of learning the desired semantics. The paper mentions that it is prohibitively
expensive to obtain human-annotated set, but can it be possible to at least obtain a
handful of real tasks to evaluate the learnt model? There are also some recent
datasets such as WikiSQL (https://github.com/salesforce/WikiSQL) that the authors
might consider in future.

Questions for the authors:

Why was MAX_VISITED only limited to 100? What happens when it is set to 10^4 or 10^6?

The Search algorithm only shows an accuracy of 0.6% with MAX_VISITED=100. What would
the performance be for a simple brute-force algorithm with a timeout of say 10 mins?

Table 3 reports an accuracy of 85.8% whereas the text mentions that the best result
is 90.1% (page 8)?

What all function names are allowed in the DSL (Figure 1)?

Can you clarify the contributions of the paper in comparison to the R3NN?

Minor typos:

page 2: allows to add constrains --> allows to add constraints
page 5: over MAX_VISITED programs has been --> over MAX_VISITED programs have been

---

> ### Author Response · Authors · 2017-12-19
> **Response to AnonReviewer2**
>
> Thanks for the detailed review and pointing out typos.
>
> Response to questions:
>
> MAX_VISITED is limited to 100 mostly due to inefficient implementation of search and DSL interpreter in Python and striving to run experiments and evaluation relatively fast. For Seq2Tree + Search it currently takes 1-4s per example or ~4 hours for full dev set. We have run model on smaller subset of dev set with MAX_VISITED 10^4 and accuracy was within the noise margin of reported, which most probably due to fact that if Seq2Tree model doesn’t know right subspace of programs, search won’t be able to recover even with large number of checks.
>
> If we let search run for 10 minutes per task, with current size of dev set it would take ~2 months to evaluate on single machine. And given there are roughly 10^25 programs with depth = 3, even 10 minutes of very optimized search would not be enough to actually find correct program of depth = 3 with brute-force search.
> (note, we consider depth = 0 just constant / function, depth = 1 a call to function with arguments, and so on. Depth = 2 already has 17,518,345,206 various programs in our DSL)
>
> Full implementation of DSL is here: https://paste.ofcode.org/SNzgEQzFAL8sVSQrrhBtRA
> We are going also add an Appendix with details on DSL. You can also find dataset here:
> https://www.dropbox.com/s/wep81pcrar5fttl/metaset3.train.jsonl.gz:
> https://www.dropbox.com/s/h3mn0abeiqy6foz/metaset3.dev.jsonl.gz
>
> Comparing our work with “Neuro-Symbolic Program Synthesis” paper:
>  * Our DSL is way more expressive: conditions, map/reduce/filter array operations, lambda functions and recursion.
>  * R3NN paper due to origin of their DSL were limited to string->string transformation. We have inputs and outputs of next types: integers, arrays of integers, strings, booleans. And we don’t have any limitations in our model to expand to any other type of inputs/outputs (for R3NN it will require to adapt IO encoder, which right now only consumes characters via LSTM).
>  * For non trivial transformations, like integer array -> boolean transformation, even 100s of examples can be not enough to understand what transformation is done, and natural language will work better. Though we agree that we should compare with just IO and IO/text combined in the encoder. Preliminary results show ~12% accuracy from just IO on our dataset for IO2SEQ.
>  * R3NN decoder iteratively adds a node to tree, thus claiming to not require an explicit search. They have also shown improving results via backtracking search. From our observations, backtracking search can get stuck in wrong space as some much earlier decision was wrong (i.e. choosing "+" with probability of 0.51 instead of "-" with 0.49). Our breadth first search in program space guided by neural model is more principled approach to do search that allows to evaluate globally most probable programs.
>
> We are working on evaluating R3NN approach on our dataset and will update here with results. Additionally, we can run our search on top of R3NN decoding strategy.
>
> Preliminary results applying IO2Seq (from Parisotto et al 2017):
> BEAM_SIZE = 1     0.02
> BEAM_SIZE = 100 0.12

---

### Official Review · AnonReviewer1 · 2017-11-27
**Decent execution, but not super new or exciting**

**Rating:** 5
**Confidence:** 4

**Review:**

This paper tackles the problem of doing program synthesis when given a problem description and a small number of input-output examples. The approach is to use a sequence-to-tree model along with an adaptation of beam search for generating tree-structured outputs. In addition, the paper assembles a template-based synthetic dataset of task descriptions and programs.  Results show that a Seq2Tree model outperforms a Seq2Seq model, that adding search to Seq2Tree improves results, and that search without any training performs worse, although the experiments assume that only a fixed number of programs are explored at test time regardless of the wall time that it takes a technique.

Strengths:

- Reasonable approach, quality is good

- The DSL is richer than that of previous related work like Balog et al. (2016).

- Results show a reasonable improvement in using a Seq2Tree model over a Seq2Seq model, which is interesting.

Weaknesses:

- There are now several papers on using a trained neural network to guide search, and this approach doesn't add too much on top of previous work. Using beam search on tree outputs is a bit of a minor contribution.

- The baselines are just minor variants of the proposed method. It would be stronger to compare against a range of different approaches to the problem, particularly given that the paper is working with a new dataset.

- Data is synthetic, and it's hard to get a sense for how difficult the presented problem is, as there are just four example problems given.

Questions:

- Why not compare against Seq2Seq + Search?

- How about comparing wall time against a traditional program synthesis technique (i.e., no machine learning), ignoring the descriptions. I would guess that an efficiently-implemented enumerative search technique could quickly explore all programs of depth 3, which makes me skeptical that Figure 4 is a fair representation of how well a non neural network-based search could do.

- Are there plans to release the dataset? Could you provide a large sample of the data at an anonymized link? I'd re-evaluate my rating after looking at the data in more detail.

---

> ### Author Response · Authors · 2017-12-19
> **Response to AnonReviewer1**
>
> Thanks for the comments!
>
> 1. Good point, here are Seq2Seq + Beam Search results. Will update paper accordingly.
>
> BEAM_SIZE = 10      0.719
> BEAM_SIZE = 100    0.728
>
> 2.  On your comment to make a more efficient enumerative search, it’s indeed limitation of our setup that our executor is in Python and limits how many programs we can find and execute in reasonable time (also reason why MAX_VISITED for beam search is relatively small to be able to evaluate on our dev set in reasonable time).
>
> For example in our DSL, given just one array as input, total number of syntactically programs of depth = 2 is 17,518,345,206. Which in our current implementation in Python took about ~4.5 hour to enumerate 1B of programs (without evaluating). It’s indeed true that more efficient implementation will be able to iterate over it may be 10-100 times faster. But for depth = 3 the total number of programs is roughly 10^25 which makes it impossible to apply regular enumerative search.
>
> Ideally, would be great to compare with traditional program synthesis techniques. But to run state-of-the-art traditional techniques (like PROSE) requires a large amount of work to build out heuristics and given our DSL is almost full LISP (with lambda functions and recursion) some things would be extremely hard to make work. I may be wrong, but my understanding is that functions like “reduce” or recursive calls can not be implemented in the PROSE’s backpropagation setup (https://microsoft.github.io/prose/documentation/prose/backpropagation/).
>
> 3.  Please find train/dev data here:
> https://www.dropbox.com/s/wep81pcrar5fttl/metaset3.train.jsonl.gz:
> https://www.dropbox.com/s/h3mn0abeiqy6foz/metaset3.dev.jsonl.gz
>
> Also our DSL full implementation can be found here: https://paste.ofcode.org/SNzgEQzFAL8sVSQrrhBtRA

---

### Official Review · AnonReviewer3 · 2017-11-27
**Paper offers a novel approach to a tricky problem and does really well**

**Rating:** 7
**Confidence:** 4

**Review:**

This paper introduces a technique for program synthesis involving a restricted grammar of problems that is beam-searched using an attentional encoder-decoder network. This work to my knowledge is the first to use a DSL closer to a full language.

The paper is very clear and easy to follow. One way it could be improved is if it were compared with another system. The results showing that guided search is a potent combination whose contribution would be made only stronger if compared with existing work.

---

### Decision · Program_Chairs · 2018-01-29
**ICLR 2018 Conference Acceptance Decision**

**Decision:**

Invite to Workshop Track

**Comment:**

the reviewers all found the problem to be important, the proposed approach to be interesting, but the manuscript to be preliminary. i agree with them.